

# Comparative analysis of nutrient composition and antioxidant activity in three dragon fruit cultivars

Afsana Yasmin[1], Mousumi Jahan Sumi[2], Keya Akter[2], Rakibul Hasan Md. Rabbi[3], Hesham S. Almoallim[4], Mohammad Javed Ansari[5], Akbar Hossain[6] and Shahin Imran[7]

[1] Department of Horticulture, Khulna Agricultural University, Khulna, Bangladesh
[2] Department of Crop Botany, Khulna Agricultural University, Khulna, Bangladesh
[3] Department of Agricultural Chemistry, Khulna Agricultural University, Khulna, Bangladesh
[4] Department of Oral and Maxillofacial Surgery, College of Dentistry, King Saud University, Riyadh, Saudi Arabia
[5] Department of Botany, Hindu College Moradabad, Mahatma Jyotiba Phule Rohilkhand University Bareilly, Moradabad, Bareilly, UP, India
[6] Soil Science Division, Bangladesh Wheat and Maize Research Institute, Dinajpur, Bangladesh
[7] Department of Agronomy, Khulna Agricultural University, Khulna, Bangladesh

Corresponding authors
Akbar Hossain,
akbarhossainwrc@gmail.com
Shahin Imran,
shahinimran124@gmail.com

## ABSTRACT

Dragon fruit has significant economic value in many countries due to has excellent nutritional content, health advantages, and adaptability to different climates, making it an important crop in the global fruit industry. This study aimed to gather comprehensive nutritional data on three dragon fruit cultivars by analysing the levels of micronutrients, fibre, carbohydrates, antioxidants, vitamins, and minerals in their pulps. Uniform dragon fruit samples underwent thorough analysis for proximate composition, mineral content, pigments, antioxidants, and vitamin C, with statistical methods used to assess significant differences among the parameters studied. The proximate composition analysis revealed significant differences among the three dragon fruit cultivars. Among the proximate components, protein ($0.40 \pm 0.02$ g/100 g), moisture ($91.33 \pm 0.88\%$), crude fibre ($0.32 \pm 0.07$ g/100 g), and ash ($1.27 \pm 0.09$ g/100 g) were more abundant in *Hylocereus costaricensis* than in *Hylocereus undatus* and *Hylocereus megalanthus*. On the other hand, *Hylocereus undatus* had higher carbohydrate ($17.02 \pm 0.63$ g/100 g) and energy ($69.74 \pm 2.44$ kcal/100 g) contents. K ($7.23 \pm 0.35$ mg/100 g), Ca ($1.61 \pm 0.13$ mg/100 g), Fe ($1.84 \pm 0.05$ mg/100 g), and Zn ($0.37 \pm 0.034$ mg/100 g) are highly abundant in *H. costaricensis*. Additionally, *Hylocereus costaricensis* had the highest anthocyanin content ($120.15 \pm 3.29$ mg/g FW) and total carotenoid content ($72.51 \pm 1.62$ mg/g FW), along with the highest vitamin C content ($8.92 \pm 0.13$ mg/g FW) and total soluble phenolic content ($572.48 \pm 20.77$ mg/100 g). Its remarkable antioxidant activity was further highlighted by the lowest $SC_{50}$ value ($13.50 \pm 0.4$ mg/mL) for its DPPH radical scavenging capacity. The total soluble sugar content was highest in *Hylocereus megalanthus* ($8.72 \pm 0.30$ g/100 g FW). Hierarchical clustering analysis revealed distinct trait and genotype associations; among the studied cultivars, *Hylocereus costaricensis* demonstrated superior performance across multiple traits. Correlation analysis indicated significant positive correlations among several traits, while principal component analysis highlighted the contribution of each trait to overall variance, with PC1 explaining 73.95% of the total variance. This study

highlights the nutritional variations among dragon fruit cultivars, with *Hylocereus costaricensis* showing superior performance, guiding dietary planning and functional food development.

# INTRODUCTION

Over the past several years, there has been a significant increase in consumers' knowledge of the importance of eating a balanced diet to prevent the development of chronic illnesses. Eating fruits and vegetables in particular has been demonstrated to play a significant impact in the prevention of many chronic diseases because plant-based meals contain a high concentration of bioactive compounds (*Mazzoni, Fernández & Capocasa, 2021*). As awareness of the health benefits of bioactive compounds has grown, there has been a recent global trend towards the usage of herbal treatments (*Ravikumar, 2014*). Furthermore, while choosing, preparing, and administering formulations to treat a range of ailments, herbal medications provide invaluable guidance. These include, among other things, diabetes, cancer, TB, and skin conditions, underscoring their diverse therapeutic potential (*Khan & Ahmad, 2019*).

The Cactaceae family encompasses climbing cactus, commonly referred to as dragon fruit (*Hylocereus* spp.), or pitaya (*Rebecca, Boyce & Chandran, 2010*; *Nurliyana et al., 2010*), which are perennial herbaceous climbers. This nonclimacteric fruit is grown in all Asian nations today and started in South and Central America (*Yusof, Adzahan & Muhammad, 2022*). Because of its unique shape and color, dragon fruit is identified as an exotic fruit (*Huynh et al., 2020*). With its high nutritional value and associated health benefits, it holds substantial economic significance across numerous countries (*Idris et al., 2013*; *Som & Wahab, 2018*). It can also withstand high temperatures and requires limited water for development and growth (*Trivellini et al., 2020*).

The entire dragon fruit, including the peel, seed, and other inedible portions, contributes significantly to the nutritional value of these byproducts (*Jiang et al., 2021*). It contains many carbs, mostly fructose, glucose, a few oligosaccharides, sugars, antioxidants such as flavonoids, hydroxycinnamates, and betalains, as well as high fibre, colour, and vitamin C contents (*Jamilah et al., 2011*; *Hossa, Numan & Cultivation, 2021*). Additionally, high amounts of several healthy compounds and minerals, including K, Ca, Zn, and Mg, can be found in dragon fruit pulp (*Tran, Yen & Ykh, 2015*). The antioxidant properties of this fruit pulp, which has free radical scavenging qualities, may be responsible for its ability to prevent disease. These compounds include alkaloids, flavonoids, vitamin C, and phenolic acids (*Gan et al., 2017*; *Pehlivan, 2017*). The prominent iron content found in red-fleshed dragon fruit increases levels of haemoglobin. The pulp provides dietary fibre for diabetics and helps reduce aortic stiffness and oxidative damage. It also assists in managing vaginal discharge and bleeding. Studies have shown that pulp also includes essential components
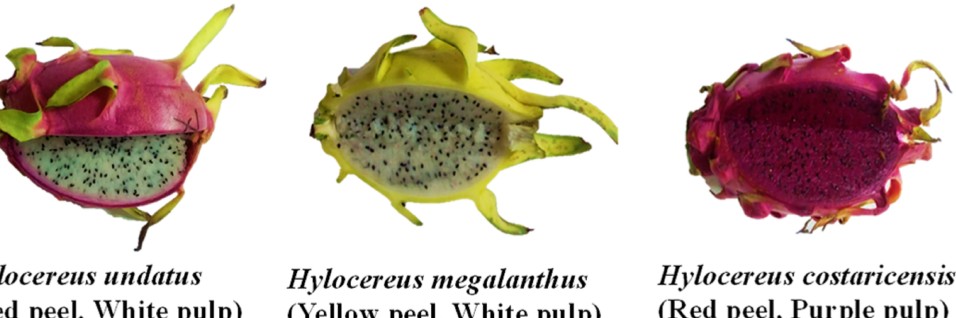

*Hylocereus undatus*
(Red peel, White pulp)

*Hylocereus megalanthus*
(Yellow peel, White pulp)

*Hylocereus costaricensis*
(Red peel, Purple pulp)

**Figure 1** Selected three species of dragon fruits (*Hylocereus* spp.) for analysis of nutrition and antioxidant.

like Mg, Zn, Fe, Ca, P, and S and essential vitamins, including C, B1, B2, B3, E, and A (*Moshfeghi et al., 2013*; *Rao & Sasanka, 2015*; *Jiang et al., 2021*).

The food processing industry uses the abundant antioxidants and pigments of dragon fruit, which include polyphenols, hydroxycinnamates, flavonoids, betacyanin, and betalains, for natural colouration and prebiotic enrichment (*Rao & Sasanka, 2015*; *Fidrianny, Ilham & Hartati, 2017*; *Huang et al., 2021*). Because of its sweet flavor, vivid color, and high nutritious content, dragon fruit has gained popularity in Bangladesh (*Patwary et al., 2013*). Farmers across the country are excited about dragon fruit growing as an emerging and promising crop with both challenges and opportunities (*Ghosh et al., 2023*).

However, despite its recognized nutritional value and health benefits, the comprehensive biochemical and nutritional profile of dragon fruit in Bangladesh remains largely unexplored. This gap in knowledge highlights the need for further research to understand the specific nutritional composition of dragon fruit varieties grown in Bangladesh. Therefore, this study aims to address this issue by gathering nutritional data on three varieties of dragon fruit. By doing so, we aim to encourage increased consumption of dragon fruit and position it as a "tropical superfruit" in the region.

## MATERIALS & METHODS

### Sample collection

Fruits of three dragon cultivars (*Hylocereus undatus*, *Hylocereus megalanthus*, and *Hylocereus costaricensis*) of uniform age (30 days after flowering) and size were collected from a dragon fruit farm located in Jhigorgacha, Jashore, and placed in Ziplock bags on 31st July. Then, the collected sample was brought to the Plant Physiology Laboratory, Department of Horticulture, Khulna University, Khulna, for further chemical analyses. For each plant species under investigation, three fruits were considered, each serving as one of the three replications (Fig. 1).

## Proximate analysis

The approximate composition of the dragon fruits in terms of carbohydrates, lipids, protein, and ash was ascertained using the technique developed by the Association of Official Agricultural Chemists (AOAC) (*Thiex, Novotny & Crawford, 2012*). The micro-Kjeldahl technique was used to determine the samples' N content. The sample undergoes digestion with sulphuric acid, followed by distillation and titration using specialized Kjeldahl apparatus to determine the nitrogen content. The nitrogen content was multiplied by 6.25 to determine the crude protein content. Weight difference methods were used to evaluate the levels of moisture and ash content, and the AOAC process with petroleum ether as a solvent was used to determine the crude fat content of dragon fruit. Soxhlet extractor was used for extracting fat content from the sample. The fat, protein, moisture, ash, crude fiber and carbohydrate were calculated using the corresponding formulae:

$$\text{Fat content (g/100 g)} = \frac{\text{(Weight of flask after extraction} - \text{Weight of flask prior to extraction)}}{\text{Weight of sample}}$$

$$\text{Nitrogen (\%)} = \frac{1.4 \times \text{acid used in titration} \times \text{normality of standard acid}}{\text{Weight of sample}}$$

$$\text{Protein (g/100 g)} = \text{N (\%)} \times 6.25$$

$$\text{Moisture (\%)} = \frac{\text{Weight of sample before drying} - \text{Weight of sample after drying}}{\text{Weight of sample before drying}} \times 100$$

$$\text{Ash (g/100 g)} = \frac{\text{Weight of ash}}{\text{Weight of sample}} \times 100$$

$$\text{Crude fiber (g/100 g)} = \frac{\text{Crude weight with fibre and ashes} - \text{Crude weight with ashes}}{\text{Weight of sample}} \times 100$$

$$\text{Total carbohydrate (\%)} = [100 - (\%\text{Protein} + \%\text{Moisture} + \%\text{Fiber} + \%\text{Fat} + \%\text{Ash})]$$

## Gross energy value

Using parameters for fat (9 Kcal/g), protein (4 Kcal/g), and carbohydrate (4 Kcal/g), the gross energy contents (Kcal/100 g samples) of purple wheat were calculated. The equation is:

$$\text{Food energy} = (\% \text{ fat content} \times 9) + (\% \text{ crude protein} \times 4) + (\% \text{ carbohydrate} \times 4)$$

## Mineral (Ca, P, Na, K, Mg, Fe, and Zn) content determination

The fruit samples were oven-dried for 72 h at 60 °C. After drying, the three dragon fruits underwent sample preparation by first being homogenized individually in a microcutter, followed by uniform mixing using homogenizer-precooled petroleum ether. After that, the mixture was filtered through a fresh muslin cloth. This procedure was carried out at least twice to produce a homogenate that was free of lipids. Subsequently, Centrifugation at 8,000 rpm for 10 min was used to further clarify each filtrate, and the precipitated material that was produced was allowed to air dry at room temperature (*Yeasmin et al., 2001*). Atomic absorption spectroscopy was used to measure the amounts of Na, K, Mg, and Zn (Perkin-Elmer, model-3110, England) in accordance with the AOAC guidelines (*Al-Mentafji, 2016*). The Fe content was estimated spectrophotometrically (Erma, AE-300) using the thiocyanate colorimetric method (*Leonard, 1990*). Ca levels were measured using the colorimetric arsenazo III method at pH 8.5 (*Bauer, 1981*). The P content in each species

was determined spectrophotometrically using the phosphovanadomolybdate method (*Leonard, 1990*).

## Anthocyanin content determination

The anthocyanin content was determined using a method involving extraction with ethanol solvent containing 0.1 M HCl followed by spectrophotometric analysis (*Ali Shehat, Sohail Akh & Alam, 2020*). The fruits were first blended to obtain a puree and then extracted with the solvent mixture. After filtration, to guarantee that the absorbance was within the spectrophotometer's linear range, the extracted solutions were diluted using pH 1.0 and pH 4.5 buffers. Absorbance readings were taken at a wavelength of 520 nm with correction at 700 nm. The molar extinction coefficient for cyanidin-3-glucoside, the dilution factor, the route length, and the difference in absorbance between pH 1.0 and pH 4.5 were all taken into account when calculating the anthocyanin concentration. (*Lee et al., 2005*).

$$\text{Anthocyanin pigment} = \frac{A.MW.DF.10^3}{\varepsilon.l}$$

where, $A = (A_{520} \text{ nm} - A_{700} \text{ nm})_{pH1.0} - (A_{520} \text{ nm} - A_{700} \text{ nm})_{pH4.5}$, $MW = 449.2 \text{ g moL}^{-1}$ for cyanidin-3-glucoside, DF = dilution factor, $l$ = Path length (cm), $\varepsilon = 26,900$ molar extinction coefficients, in L/mol cm for cyanidin-3-glucoside and $10^3$ = Factor for conversion from g to mg

## Pigment determination

Total carotenoids were determined using a modified procedure from *Lichtenthaler (1987)*. To determine the concentration of carotenoids in the fruit sample, 1 gram of fully blended and finely chopped fruit tissue was placed in a sanitized mortar. After the tissue was finely pulped, 20 ml of 80% acetone was added. After 5 minutes of centrifuging the mixture at 5,000 rpm, the supernatant was moved to a 100 mL volumetric flask. The grinding and centrifugation steps were repeated with fresh portions of 80% acetone until the residue became colorless. Then, using 80% acetone, the volume was adjusted to 100 mL. Subsequently, the solution's absorbance was measured against a blank at 510 and 480 nm. Finally, the following formula was used to determine the concentration of carotenoids (mg/g tissue):

$$\text{Carotenoids (mg/g tissue)} = 7.6(A_{480}) - 1.49(A_{510}) \times \frac{V}{W \times 10}$$

where 'A' represents the absorbance at the specified wavelengths, 'V' is the final volume of the carotenoid solution in 80% acetone and 'W' is the fresh weight of the extracted tissue.

## Total phenolic content assay

A modified approach from was used to measure the total phenolic compounds (*Albano & Miguel, 2011*). The methanolic extract was obtained in a 1.5 µL tube following a 30-minute dark period and a 5-minute centrifugation at 15,000 rpm. Subsequently, the phenolic content was ascertained using the supernatant. The standard used to calculate the total phenolic content was gallic acid. Plant extracts totaling 330 µL were put to a 50 mL test tube. After that, the tube was filled with three mL of 10% $Na_2CO_3$ solution and 16 µL of Folin-Ciocalteu reagent. After that, the mixture was allowed to sit at room temperature for 30 min in the dark. The compounds' total phenol content was then ascertained by measuring the absorbance at 760 nm.

## Total flavonoid content determination

Using a modified gravimetric technique, the flavonoid content was found (*Harborne, 1973*). For the determination of flavonoids, 5 grams of the sample was finely crushed and blended with 50 mL of 80% methanol. This combination was subsequently extracted for 10-hours in a water bath at 40 °C. After extraction, to get rid of any solid particles, the solution was filtered using 125 mm filter paper. Following that, the filtrate was moved to a crucible and dried over a water bath at room temperature. Finally, the dry residue, representing the flavonoids, was weighed to determine the final quantity.

## Vitamin C content determination

The principle behind the tritimetric estimation of vitamin C involves the use of a dye solution, 2,6-dichlorophenol indophenol, which exhibits a colour change from blue to red in the presence of ascorbic acid. This reaction is specific and quantitative for ascorbic acid in the concentration range of 10–35 µg/ml. The reagents used included meta-phosphoric acid and the indophenol dye solution. The extraction of a known weight tissue sample with 3% meta-phosphoric acid was performed, followed by diluting to a specified volume. Then, the endpoint of the reaction was reached when an aliquot of this solution was titrated with the indophenol dye solution and a persistent pink color appeared. The dye factor was then ascertained after the dye solution had been normalized using a standard ascorbic acid solution. Ultimately, the following formula was used to determine the vitamin C content. Following *Xiao et al. (2012)*, this procedure was changed.

$$\text{Vitamin C (mg/100 g FW)} = \frac{e \times d \times b}{c \times a}$$

where, a = weight of sample, b = volume made with metaphosphoric acid, c = volume of aliquot taken for estimation, d = dye factor and e = average burette reading for sample

## DPPH radical scavenging capacity assay

The stability of the 2,2-diphenyl-l-picrylhydrazyl radical (DPPH) was assessed using thin layer chromatography (TLC) (*Cieśla et al., 2012*). Initially, 21 clean test tubes were prepared, with 9 designated for varying plant extract concentrations (between 2 and 512 µg/ml), 9 for different concentrations of ascorbic acid (also between 2 and 512 µg/ml), and one for the blank solution. Both the plant extract and ascorbic acid were dissolved in ethanol to form stock solutions, from which dilutions were prepared for each concentration. Additionally, A 0.004% DPPH (2,2-diphenyl-1-picrylhydrazyl) solution was made with ethanol. Subsequently, two mL of each concentration of the plant extract and in separate test tubes, ascorbic acid and six mL of the DPPH solution were combined, and the tubes were then allowed to sit in the dark at room temperature for half an hour. A blank solution containing only ethanol and DPPH was also prepared. After the incubation period, A UV spectrophotometer was used to measure each test tube's absorbance at 517 nm. The experiment also included the preparation of a standard solution of ascorbic acid at a dosage of 10 mg/10 ml in ethanol for calibration purposes. The percentage of scavenging activity was determined as follows:

$$\text{SC}_{50} = \frac{Ac - As}{Ac} \times 100$$

To calculate the $SC_{50}$ value, plot the percentage of radical scavenging activity against the extract concentration, where 'As' represents the sample absorbance and 'Ac' represents the control absorbance (no extract).

## Statistical analysis

To find out whether there were any significant differences between the groups, Minitab 17.3 was used to do a two-way ANOVA on the mean values of all the parameters examined for all the species. The Tukey HSD test ($p < 0.05$) was applied to the means in the event of a significant F-*ratio* to identify any significant differences between the mean values. "Corrplot" package was used for correlation analysis. To construct a heatmap in R 4.3.2, the "heatmap.2" package was used. In order to conduct principal component analysis (PCA), the "GGally" and "factoextra" packages were used.

## RESULTS

### Proximate composition of three dragon fruit cultivars

Table 1 provides a comparative analysis of the proximate composition of three dragon fruit cultivars: *Hylocereus undatus*, *Hylocereus megalanthus*, and *Hylocereus costaricensis*. The data indicate significant differences in various nutrient components among the cultivars. Although there were no significant differences among the three cultivars, *Hylocereus undatus* exhibited the highest carbohydrate content at $17.02 \pm 0.63$ g/100 g, followed by *Hylocereus megalanthus* at $15.76 \pm 1.05$ g/100 g, and *Hylocereus costaricensis* had the lowest carbohydrate content at $6.61 \pm 1.03$ g/100 g (Table 1). The species with the highest protein content was *Hylocereus costaricensis* ($0.40 \pm 0.02$ g/100 g), while *Hylocereus undatus* had the lowest ($0.22 \pm 0.02$ g/100 g) (Table 1). The protein content of the *Hylocereus costaricensis* and *Hylocereus megalanthus* cultivars did not differ much. The fat content remained relatively consistent across all the cultivars, with minor variations observed. Similar to the carbohydrate content, *Hylocereus costaricensis* had the highest moisture content ($91.33 \pm 0.88\%$), while the other two cultivars had lower levels. Among the three cultivars, *Hylocereus costaricensis* exhibited the maximum amount of crude fiber ($0.32 \pm 0.07$ g/100 g) and ash content ($1.27 \pm 0.09$ g/100 g) (Table 1). *Hylocereus megalanthus* had the lowest values, and *Hylocereus undatus* had the lowest crude fibre ($0.07 \pm 0.01$ g/100 g) and ash ($0.60 \pm 0.06$ g/100 g) contents. In terms of energy content, *Hylocereus megalanthus* had the highest energy content, at $64.97 \pm 4.25$ kcal/100 g, while *Hylocereus costaricensis* had the lowest amount of energy, at $28.68 \pm 4.07$ kcal (Table 1).

### Mineral composition of the three dragon fruit cultivars

The mineral content differed significantly among the three cultivars examined (Table 2). *Hylocereus costaricensis* had the highest K content ($7.23 \pm 0.35$ mg/100 g), while *Hylocereus megalanthus* and *Hylocereus undatus* had the lowest K content. Similarly, *compared with the other two cultivars, H. costaricensis* had the highest calcium ($1.61 \pm 0.13$ mg/100 g), iron ($1.84 \pm 0.05$ mg/100 g), and zinc ($0.37 \pm 0.034$ mg/100 g) contents. *Hylocereus costaricensis* had the highest magnesium ($9.58 \pm 0.42$ mg/100 g) and phosphorus ($5.67 \pm 0.054$ mg/100 g) contents, whereas with the most sodium, *Hylocereus megalanthus* was the most

**Table 1 Proximate composition of three tested dragon fruit cultivars (dry weight basis).** The data represent the standard error of the mean (SEM, $n = 3$). Mean values do not share a common letter in each parameter are significantly different from each other at a 5% level of probability

| Components | *Hylocereus undatus* | *Hylocereus megalanthus* | *Hylocereus costaricensis* |
|---|---|---|---|
| Carbohydrate (g/100 g) | $17.02 \pm 0.63^a$ | $15.76 \pm 1.05^a$ | $6.61 \pm 1.03^a$ |
| Protein (g/100 g) | $0.22 \pm 0.02^b$ | $0.35 \pm 0.04^a$ | $0.40 \pm 0.02^a$ |
| Fat (g/100 g) | $0.09 \pm 0.01^a$ | $0.06 \pm 0.01^a$ | $0.07 \pm 0.02^a$ |
| Moisture (%) | $82.00 \pm 0.58^b$ | $83.00 \pm 1.15^b$ | $91.33 \pm 0.88^a$ |
| Crude fiber (g/100 g) | $0.07 \pm 0.01^b$ | $0.13 \pm 0.02^{ab}$ | $0.32 \pm 0.07^a$ |
| Ash (g/100 g) | $0.60 \pm 0.06^b$ | $0.70 \pm 0.12^b$ | $1.27 \pm 0.09^a$ |
| Energy (Kcal/100 g) | $69.74 \pm 2.44^a$ | $64.97 \pm 4.25^a$ | $28.68 \pm 4.07^b$ |

**Table 2 Mineral composition of tested three dragon fruit cultivars (dry weight basis).** The data represent the standard error of the mean (SEM, $n = 3$). Mean values do not share a common letter in each parameter are significantly different from each other at a 5% level of probability

| Components | *Hylocereus undatus* | *Hylocereus megalanthus* | *Hylocereus costaricensis* |
|---|---|---|---|
| K (mg/100 g) | $3.39 \pm 0.21^b$ | $4.32 \pm 0.18^b$ | $7.23 \pm 0.35^a$ |
| Ca (mg/100 g) | $0.92 \pm 0.03^b$ | $1.21 \pm 0.06^b$ | $1.61 \pm 0.13^a$ |
| Fe (mg/100 g) | $0.75 \pm 0.02^b$ | $0.86 \pm 0.04^b$ | $1.84 \pm 0.05^a$ |
| Mg (mg/100 g) | $8.54 \pm 0.36^a$ | $6.46 \pm 0.34^b$ | $9.58 \pm 0.42^a$ |
| Na (mg/100 g) | $15.49 \pm 0.72^c$ | $24.69 \pm 0.043^a$ | $17.81 \pm 0.34^b$ |
| P (mg/100 g) | $4.14 \pm 0.043^c$ | $6.66 \pm 0.31^a$ | $5.67 \pm 0.054^b$ |
| Zn (mg/100 g) | $0.18 \pm 0.006^b$ | $0.24 \pm 0.022^b$ | $0.37 \pm 0.034^a$ |

($24.69 \pm 0.043$ mg/100 g) among the three cultivars (Table 2). Conversely, *Hylocereus undatus* generally has lower levels of these minerals than the other cultivars. Significant differences are denoted by letters (a, b, c), with different letters indicating statistical significance (Table 2).

## Anthocyanin content in fruits of three different dragon fruit cultivars

Among the three dragon fruit cultivars studied, the anthocyanin concentration varied greatly, ranging from $120.15 \pm 3.29$ to $12.67 \pm 0.16$ mg/g FW. *Hylocereus undatus* fruit ($18.51 \pm 0.76$ mg/g FW) and *Hylocereus costaricensis* fruit ($120.15 \pm 3.29$ mg/g FW) exhibited the highest anthocyanin concentration. The plant *Hylocereus megalanthus* had the lowest anthocyanin concentration ($12.67 \pm 0.16$ mg/g FW). Notably, the anthocyanin content did not significantly differ between the *Hylocereus undatus* and *Hylocereus megalanthus* dragon fruit cultivars (Fig. 2A).

## Total carotenoid content in fruits of three different dragon fruit cultivars

The carotenoid content of the three different dragon fruit cultivars varied significantly. The carotenoid content ranged from $72.51 \pm 1.62$ to $13.53 \pm 1.88$ mg/g FW, as shown in Fig. 2B. *Compared with the other cultivars studied, H. costaricensis had the highest total carotenoid content ($72.51 \pm 1.62$ mg/g FW). Hylocereus undatus* had a lower carotenoid content ($15.89 \pm 0.65$ mg/g FW) than the other species but had the second lowest carotenoid content.

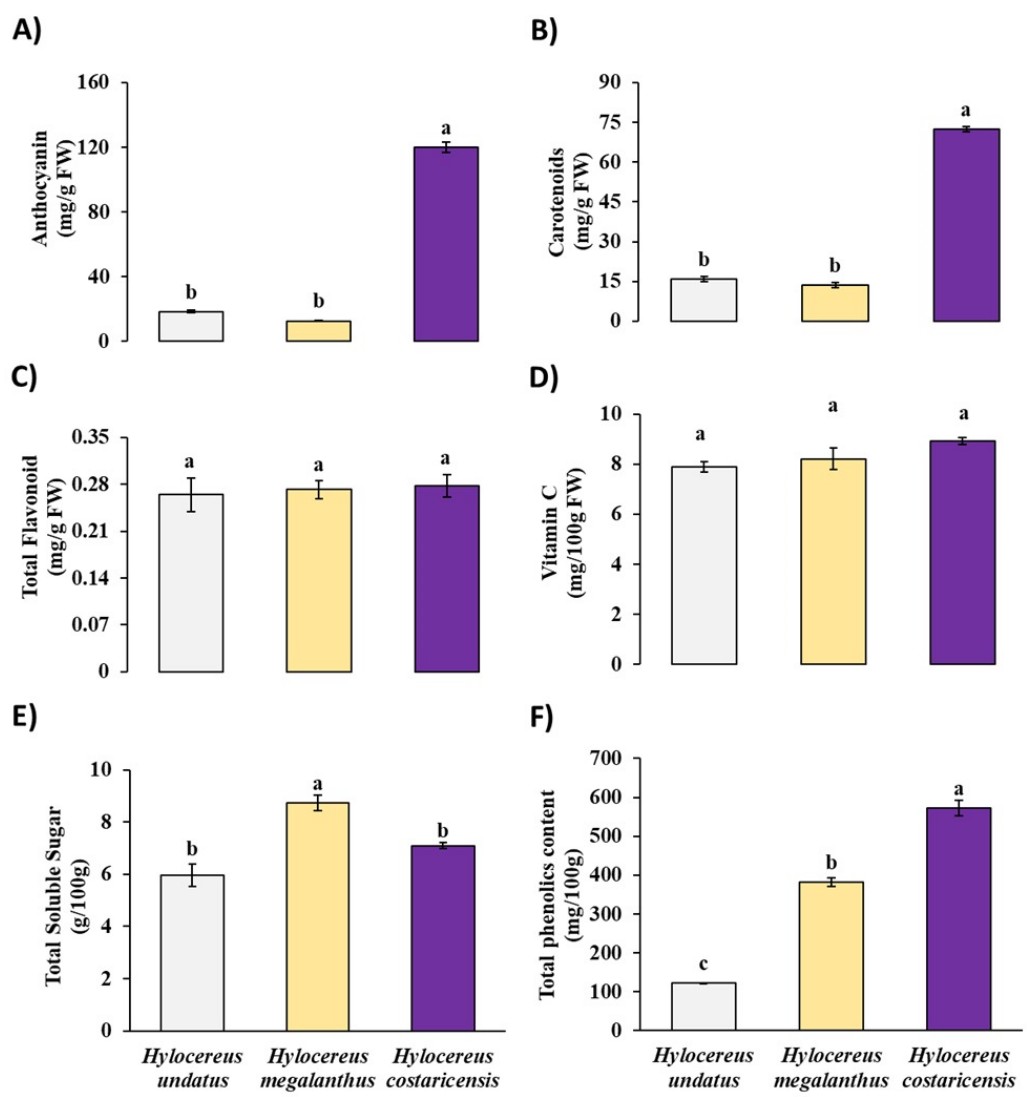

**Figure 2** **Content of (A) anthocyanin, (B) total carotenoids, (C) total soluble flavonoids, (D) vitamin C, (E) total soluble sugar, (F) total phenolics in three dragon fruit cultivars.** The error bars above each bar in the graph represent the standard error of the mean (SEM, $n = 3$). Mean values do not share a common letter in each parameter are significantly different from each other at a 5% level of probability.

Among the cultivars, *Hylocereus megalanthus* had the lowest carotenoid concentration, at 13.53 ± 1.88 mg/g FW (Fig. 2B). However, the total carotenoid concentration of the *Hylocereus megalanthus* and *Hylocereus undatus* dragon fruit cultivars did not significantly differ in terms of anthocyanin content.

## Total soluble flavonoid content in fruits of three different dragon fruit cultivars

Notably, the flavonoid content of the three dragon fruit cultivars did not differ much (Fig. 2C). On the other hand, *Hylocereus megalanthus* (0.27 ± 0.014 mg/g FW) and *Hylocereus costaricensis* (0.28 ± 0.016 mg/g FW) had the highest total soluble flavonoid

concentration. *Hylocereus undatus* fruit has the lowest amount of soluble flavonoids (0.26 ± 0.025 mg/g FW) (Fig. 2C).

### Vitamin C content in fruits of three different dragon fruit cultivars

The investigation of the vitamin C content across three dragon fruit cultivars revealed a range from 8.92 ± 0.13 to 7.89 ± 0.21 mg/g FW (Fig. 2D). Notably, *Hylocereus costaricensis* exhibited the highest vitamin C content at 8.92 ± 0.13 mg/g FW, followed closely by *Hylocereus megalanthus* at 8.21 ± 0.43 mg/g FW. Conversely, *Hylocereus undatus* had the lowest vitamin C content among the cultivars, at 7.89 ± 0.21 mg/g FW (Fig. 2D). Notably, The total soluble flavonoid content and the vitamin C content of the three dragon fruit cultivars did not significantly differ from one another.

### Total soluble sugar content in fruits of three different dragon fruit cultivars

The total soluble sugar content averaged 5.96 ± 0.42 g/100 g FW for *Hylocereus undatus* fruit, 8.72 ± 0.30 g/100 g FW for *Hylocereus megalanthus* fruit, and 7.10 ± 0.11 g/100 g FW for *Hylocereus costaricensis* fruit (Fig. 2E). Significantly, the *Hylocereus megalanthus* variety had the highest soluble sugar content compared to the other two varieties. However, there were no discernible variations seen in the soluble sugar concentrations of the *Hylocereus costaricensis* and *Hylocereus undatus* fruits.

### Total phenolic content in fruits of three different dragon fruit cultivars

The average total soluble phenolic content was 122.7 ± 1.23 mg/ 100× g for *Hylocereus undatus* fruit, 381.9 ± 10.91 mg/ 100× g for *Hylocereus megalanthus* fruit, and 572.48 ± 20.77 mg/ 100× g for *Hylocereus costaricensis* fruit. These values represent the concentration of phenolic compounds, and they are present in every dragon fruit species and are renowned for their antioxidant qualities. Overall, The *Hylocereus costaricensis* variety exhibited the greatest level of total soluble phenolic content, with the *Hylocereus megalanthus* and *Hylocereus undatus* variants following suit (Fig. 2F).

### $SC_{50}$ to scavenge DPPH ($\mu$g/mL) in fruits of three different dragon fruit cultivars

The three studied dragon fruit cultivars have greatly different $SC_{50}$ values, which indicate the quantity of antioxidant material needed to scavenge 50% of the DPPH radical in the assay system (Fig. 3). The concentration ranged from 13.50 ± 0.4 mg/mL for *Hylocereus costaricensis* to 74.92 ± 1.43 mg/mL for *Hylocereus megalanthus* fruit (Fig. 2). *Hylocereus undatus* had the second greatest effect (55.04 ± 2.15 mg/g FW) (Table 1).

### Hierarchical clustering and coclustering analysis

Figure 4A shows a hierarchical clustering heatmap of fruits from three dragon fruit species based on their proximate composition, mineral matter content, and several important antioxidant properties. Each row cluster represents the variability of each of the three species. Cluster 1 outperformed the remaining two-row clusters (Fig. 4B). Again, each of the six column clusters contained a distinct set of traits. Cluster 3 contained only one trait, whereas Clusters 1 and 4 contained two traits each. Cluster 3 and Cluster 5 had five and

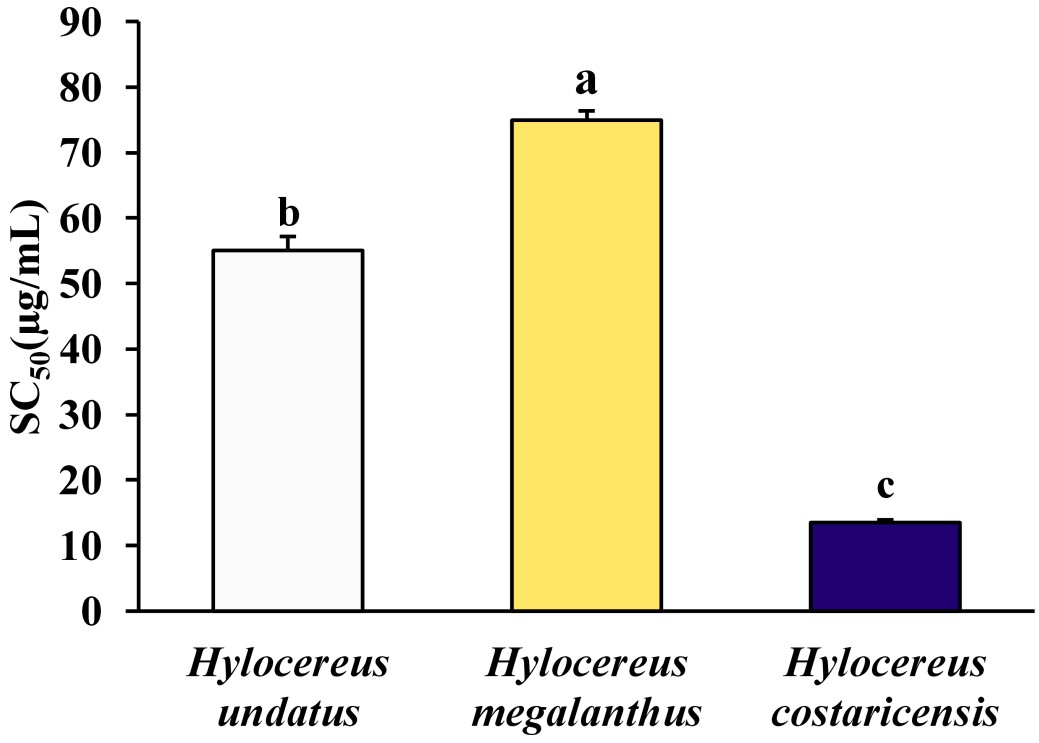

**Figure 3** **The SC$_{50}$ values for DPPH free radical scavenging ($\mu$g/mL FW) in three dragon fruit cultivars.** The error bars above each bar in the graph indicate the standard error of the mean (SEM, $n = 3$). Mean values that do not share a common letter in each parameter differ significantly from one another at the 5% level of probability.

three traits, respectively. Surprisingly, Cluster 2 included the eight most closely related traits (Fig. 4A).

## Correlation analysis

The correlation coefficients of the twenty-one qualities under study indicated the degree of relationship between them, as shown in Fig. 5. All the studied parameters were significantly correlated. There is a strong positive correlation observed for particular traits, *i.e.,* Pr, Fl, Tp, Ac, As, MS, Fe, Ca, Vc, Zn, CF and K. This suggests that when considering any combination of two traits from the mentioned set, there is a tendency for an increase in the value of one trait to coincide with an increase in the value of the other trait. On the other hand, a decrease in the Na value is accompanied by a decrease in the Ft and Mg values. The same result is found for the TS values with respect to the Ft and Mg values. However, it was not significantly correlated with any of the other studied traits except for a negative correlation with Ft and Mg. Again, IC, Cb, En and Ft mostly showed negative correlations with all the other studied traits except for one or two exceptions (Fig. 5).

## Principal component analysis

To find out if genotypic behavior can be explained in more detail by creating two new variables (PC1 and PC2) that incorporate the original variables/traits to differing degrees,
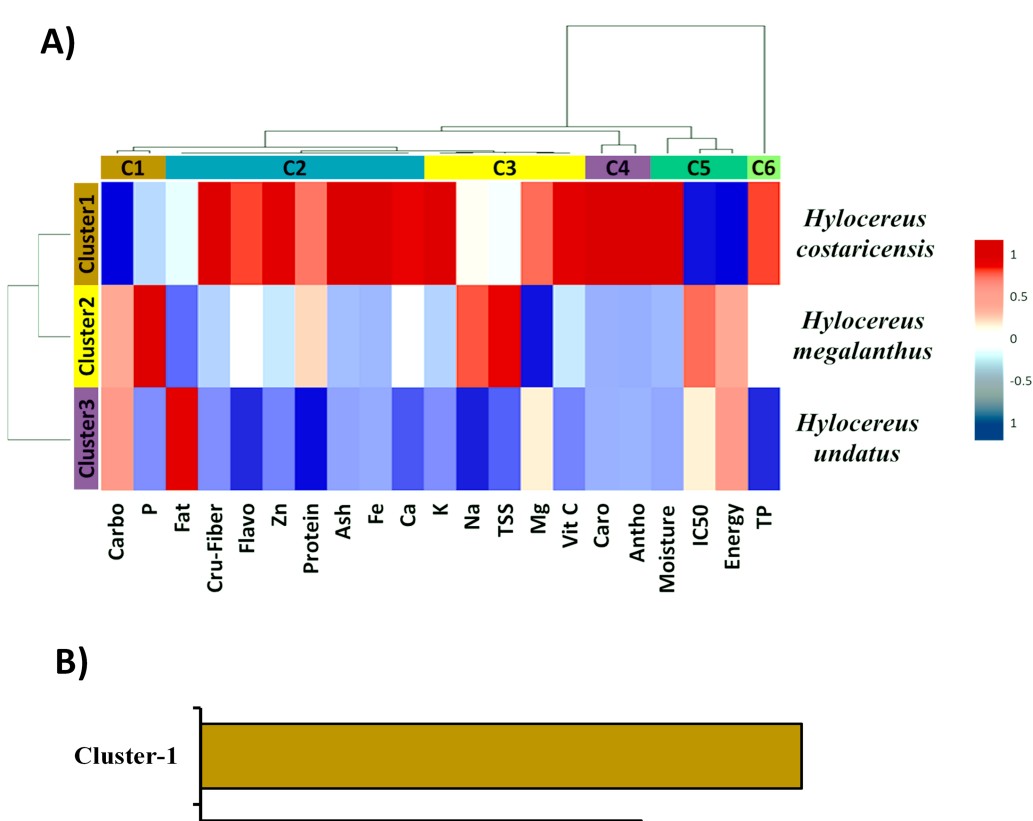

**Figure 4** **(A) Heatmap with a clustering approach and (B) comparative row cluster analysis for the studied twenty-one traits.** The studied parameters were carbohydrate (Carbo); Phosphorus (P); fat; crude fiber (Cru-fiber); flavonoid (Flavo); Zinc (Zn); protein; ash; Iron (Fe); calcium (Ca); potassium (K); Sodium (Na); total soluble sugar (TSS); magnesium (Mg); vitamin C (Vit C); carotenoids (Caro); anthocyanin (Antho); moisture; DPPH radical scavenging capacity (IC50); Energy; total phenolics (TP).

principal component analysis (PCA), a sort of multivariate analysis, was used (Fig. 6). The retrieved eigenvalues for the PCs in this investigation ranged from 5.47 (PC2) to 15.53 (PC1), all of which were more than one. The values signify the amount of variance captured by each principal component, with PC1 having the highest value of 15.53, followed by 5.47 for PC2, and an extremely low value for PC3 (Table 3). The percentage of total variance that each primary component accounts for is shown by the explained variance, with PC1 explaining 73.95%, PC2 explaining 26.05%, and PC3 being negligible. According to the cumulative variance percentage, PC1 accounts for 73.95% of the variation overall, whereas PC1 and PC2 combined account for 100% (Fig. 6). The eigenvectors represent

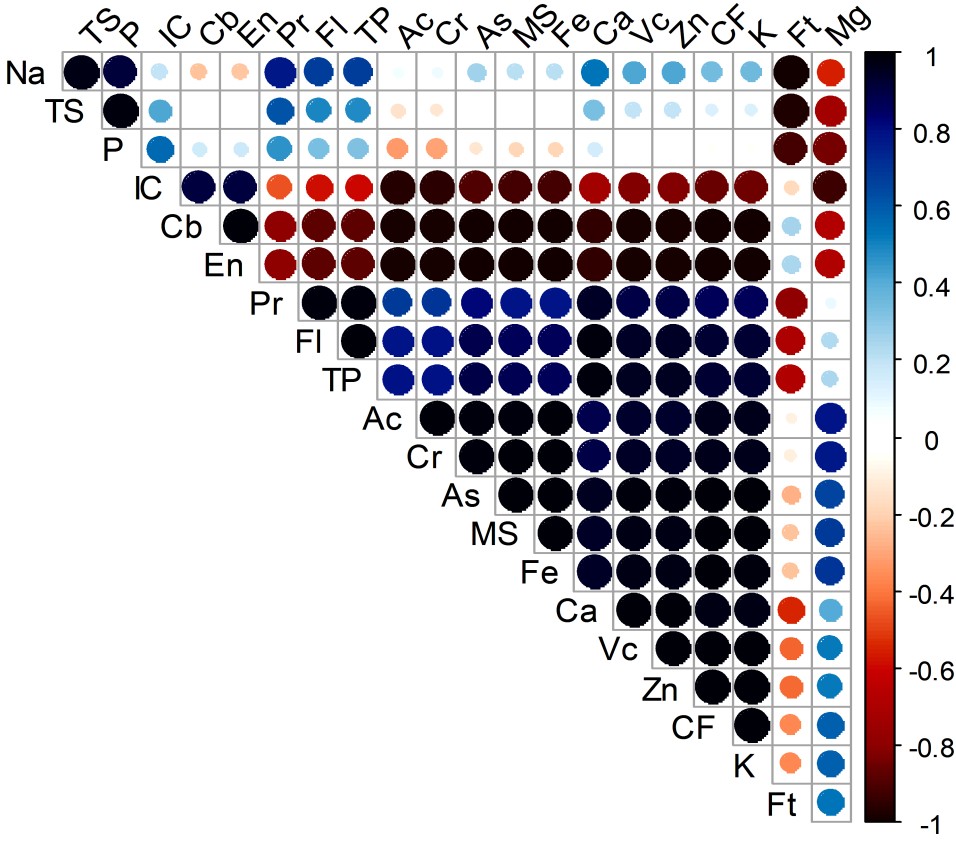

**Figure 5 Correlation matrix analysis of the studied parameters of three dragon fruit species.** Here, Cb, carbohydrate; P, Phosphorus; CF, crude fiber; Fl, flavonoid; Zn, Zinc; Fe, Iron; Ca, Calcium; K, Potassium; Na, Sodium; TS, total soluble sugar; Mg, Magnesium; Vc, vitamin C; As, ash; MS, moisture; Cr, carotenoids; Ac, anthocyanin; En, energy; Pr, protein; IC, DPPH radical scavenging capacity, TP, total phenolics; TS, Total soluble sugar.

the coefficients of the original variables in the linear combination defining each principal component, demonstrating their contributions to each component's variance.

## DISCUSSION

The comparative analysis provided by the results underscores the nutritional diversity among the examined dragon fruit cultivars. The nutritional diversity among dragon fruit cultivars informs dietary choices and product development to meet varying consumer needs (*Priya & Alur, 2023*; *Turton et al., 2023*). Variances in carbohydrate and protein levels are particularly relevant for individuals managing diabetes or seeking protein-rich options (*Kaufman et al., 2023*). However, *Hylocereus costaricensis,* which has the highest moisture content among cultivars, is relevant for food processing and for consumers to value juiciness. The mineral content of fruits is of significant interest due to its potential health implications and dietary considerations. Our findings revealed significant differences in mineral content among the three examined cultivars. Notably, Comparing *Hylocereus costaricensis* to the other two cultivars, the latter had the highest amounts of various

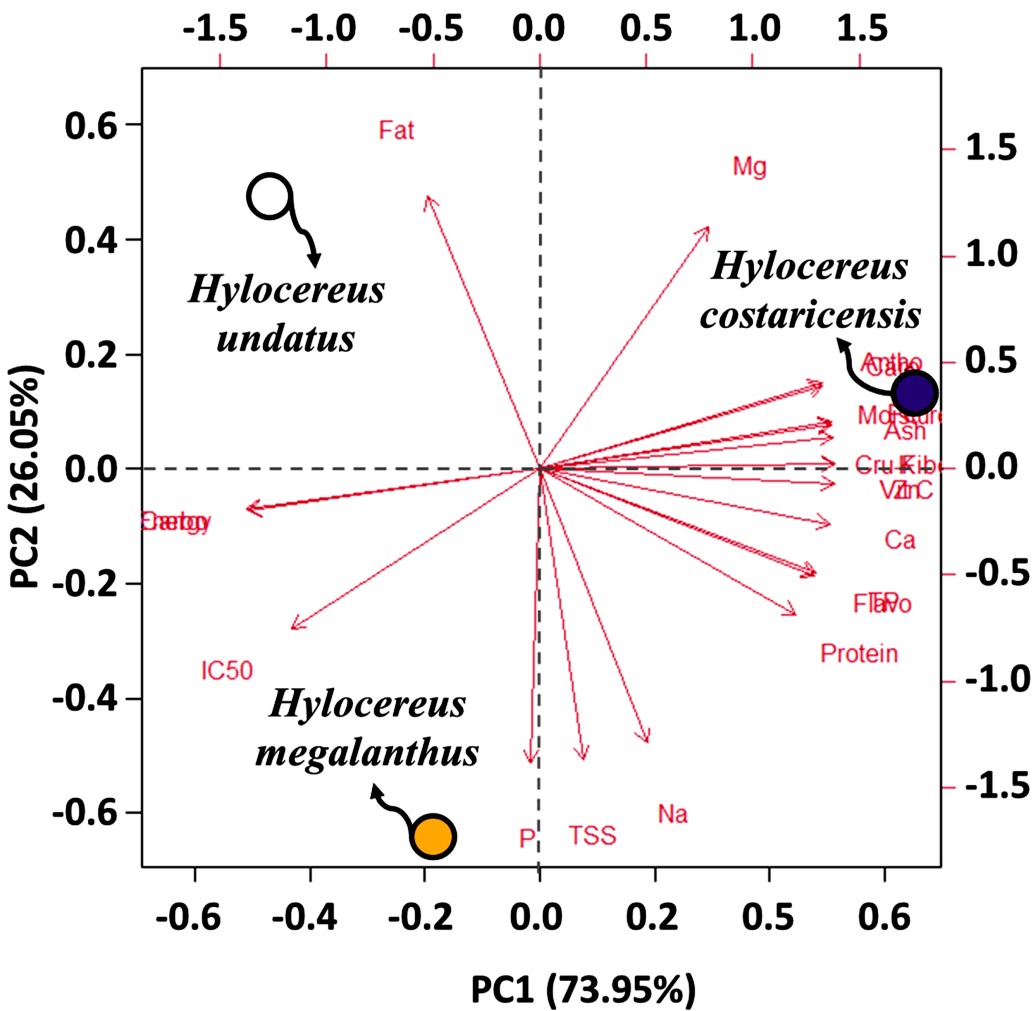

**Figure 6  PCA analysis was done based on all of the studied parameters.** The studied parameters were carbohydrate (Carbo); Phosphorus (P); fat; crude fiber (Cru-fiber); flavonoid (Flavo); Zinc (Zn); protein; ash; Iron (Fe); calcium (Ca): potassium (K); Sodium (Na); total soluble sugar (TSS); magnesium (Mg); vitamin C (Vit C); carotenoids (Caro); anthocyanin (Antho); moisture; DPPH radical scavenging capacity (IC50); Energy; total phenolics (TP).

important minerals. Specifically, this cultivar demonstrated significantly elevated levels of K, Ca, Fe, Zn, Mg, and P compared to *Hylocereus megalanthus* and *Hylocereus undatus*. Studies by *Arivalagan et al. (2021)* and *Nur et al. (2023)* also showed a similar outcome. Potassium is a vital mineral that is involved in many physiological functions, such as neuron activity, muscular contraction, and fluid equilibrium (*Udensi & Tchounwou, 2017*). The significantly higher K content observed in the abovementioned cultivar suggests that consuming this variety may contribute more effectively to meeting daily potassium requirements compared to the other cultivars. Similarly, the higher concentrations of Ca, Mg, P, Fe, and zinc in *Hylocereus costaricensis* are noteworthy, as these minerals play vital roles in bone health, energy metabolism, DNA synthesis, oxygen transport, and immune function, respectively (*Udensi & Tchounwou, 2017*; *Mitra et al., 2022*). The findings of this

**Table 3** Extracted eigenvalues and latent vectors of three dragon fruit cultivars associated with the first three principal components.

| Variable | Principal Components | | |
| --- | --- | --- | --- |
| | PC1 | PC2 | PC3 |
| Extracted eigenvalues | 15.53 | 5.47 | 3.96e−30 |
| Explained variance (%) | 73.95 | 26.05 | 1.89e−29 |
| Cumulative variance (%) | 73.95 | 100 | 100 |
| *Traits* | *Eigen vectors* | | |
| Antho | 0.242616 | 0.125286 | 0.121834 |
| Caro | 0.243563 | 0.119952 | −0.04736 |
| Flavo | 0.236538 | −0.15481 | 0.005196 |
| Vit C | 0.2534 | −0.02231 | 0.703758 |
| IC | −0.21337 | −0.23142 | −0.01385 |
| Carbo | −0.25136 | −0.05861 | −0.05767 |
| Protein | 0.220575 | −0.21138 | −0.14363 |
| Fat | −0.09704 | 0.395076 | −0.02613 |
| Cru Fiber | 0.253715 | 0.00744 | 0.063259 |
| Moisture | 0.250854 | 0.064454 | −0.56897 |
| Ash | 0.252233 | 0.046737 | 0.002084 |
| Energy | −0.25117 | −0.06081 | 0.056391 |
| K | 0.037521 | −0.42287 | 0.00647 |
| Ca | 0.237271 | −0.15159 | −0.07577 |
| Fe | −0.00763 | −0.42738 | −0.03435 |
| Mg | 0.092484 | −0.39816 | −0.25419 |
| TSS | 0.253417 | −0.02201 | 0.047493 |
| TP | 0.253726 | 0.006325 | 0.027672 |
| P | 0.249268 | −0.08004 | −0.18296 |
| Na | 0.250599 | 0.067213 | −0.13322 |
| Zn | 0.144956 | 0.350944 | 0.077736 |

investigation align with those of *Singh et al. (2022)* where there was an increasing trend in all the minerals up to a certain period, except for phosphorus content, which increased until the last stage of evaluation. Anthocyanins, phenolic compounds, offer health benefits, including cardiovascular disease and cancer prevention (*Diaconeasa et al., 2020*). They determine the colors of fruits and flowers, influencing consumer preference and market value, with variations attributed to gene expression levels (*Khoo et al., 2017*; *Zheng et al., 2019*). Interestingly, the fruit of *Hylocereus costaricensis* had the highest anthocyanin content in the current investigation, possibly because of the direct involvement of this variety in the anthocyanin biosynthetic pathway compared to those of the other two cultivars. Some external factors, such as temperature, might influence anthocyanin accumulation in the studied cultivars. *Nissim-Levi et al. (2007)* reported that low temperatures increase, and elevated temperatures decrease pigment concentrations in some fruits.

Generally, carotenoids protect plants from photooxidation (*Saini, Nile & Park, 2015*). Additionally, dietary carotenoids can be converted by humans into physiologically active vitamin A (*Abirami et al., 2021*). The total carotenoid concentration of *Hylocereus*

*costaricensis* was found to be about 80 times higher in this study than in *Hylocereus undatus* and *Hylocereus megalanthus*, the other two cultivars. Similarly, pro-vitamin A may be found in abundance in dragon fruits with a high carotenoid concentration, according to a different study by *Trumbo et al. (2001)*. Another study identified high concentrations of lutein, β-carotene, and vitamin A in *Hylocereus* spp., emphasizing its nutritional value (*Moo-Huchin et al., 2017*). Therefore, this particular cultivar, *Hylocereus costaricensis* may be utilized in the creation of nutraceutical goods to lessen the prevalence of vitamin A insufficiency in people. The high concentration of plant pigment chlorophyll, which is produced in chloroplasts and gives plants their various colors, may be the cause of the high carotenoid content in that specific cultivar (*Cazzaniga et al., 2016*).

When compared to other fruits, dragon fruit's high flavonoid content makes it a potentially useful source of polyphenols for human consumption (*Lako, Trenerry & Rochfort, 2008*). Flavonoids, which are abundant secondary metabolites, not only contribute to plant color but also offer various physiological benefits (*Muhaisen, 2014*). Again, the total flavonoids or groups safeguard the circulatory system in addition to having diabetic medications, antiobesity, and anticancer effects (*Ballard & Maróstica, 2019*). The current study found that the pulp of dragon fruit has varying levels of flavonoids, ranging from low to medium levels of *Hylocereus undatus* and *Hylocereus megalanthus*, as well as from medium to high levels of *Hylocereus costaricensis*. Similar results for flavonoid content were reported by *Ramli, Ismail & Rahmat (2014)*.

Regular consumption of dragon fruit, which has a high vitamin C content, improves wound healing properties and speeds up the healing of cut areas. It also strengthens the immune system and encourages the body's other antioxidants to function (*Cheah et al., 2016*). *Hylocereus costaricensis* showed the highest concentration of vitamin C of all the cultivars, indicating that this cultivar may be more nutritious than the others when it comes to vitamin C content.

These observed disparities in vitamin C content among the studied dragon fruit cultivars could stem from various factors, including genetic differences, environmental conditions, cultivation practices, and postharvest handling (*Martínez, Valenzuela & Jamilena, 2021*; *Shah et al., 2023*).

A fruit's taste, sweetness, and general consumer attractiveness are all significantly influenced by the amount of soluble sugar it contains overall. The fruits of the three cultivars under examination, especially *Hylocereus megalanthus,* have a higher total soluble sugar content than the fruits of the other two. This suggests that the *Hylocereus megalanthus* cultivar may offer a sweeter taste experience, potentially appealing to consumers who prefer sweeter fruits. Conversely, the *Hylocereus undatus* and *Hylocereus costaricensis* displayed similar levels of soluble sugars, with no significant differences between them. Which indicates that, variations in soluble sugar content among dragon fruit cultivars stem from factors such as genetic disparities, environmental conditions, and ripeness at harvest (*Magalhães et al., 2019*). Environmental factors such as soil composition, temperature, and sunlight exposure may impact sugar synthesis and accumulation in these cultivars, according to *Wakchaure et al. (2023)*, who also noticed a comparable reaction in cultivars of dragon fruit.

The DPPH assay assesses antioxidant activity by measuring the reduction of DPPH radicals in solution, which is commonly used for evaluating compounds (*Baliyan et al., 2022*). The concentration required to scavenge 50% of the DPPH radical is shown by the $SC_{50}$ figure, serves as a critical indicator of antioxidant potency (*Baliyan et al., 2022*). Among the three dragon fruit varieties, *Hylocereus costaricensis* displayed the highest antioxidant activity, with *Hylocereus megalanthus* requiring a relatively high concentration for comparable DPPH radical scavenging, while *Hylocereus undatus* showed intermediate activity. *Mahdi et al. (2018)* and *Zain, Nazeri & Azman (2019)* also carried out a study of a similar nature.

The diversity in antioxidant potential among dragon fruit varieties underscores the influence of factors such as genetics, environment, and ripeness stages (*Abirami et al., 2021*; *Shah et al., 2023*). These variations have implications for selecting suitable varieties for maximizing antioxidant benefits, with potential applications in food, pharmaceuticals, and natural antioxidants (*Jiang et al., 2021*; *Nishikito et al., 2023*).The study thoroughly examines the nutritional makeup of many varieties of dragon fruit, identifying notable variations in antioxidants, flavonoids, and minerals. These results provide new information on product development, dietary choices, and possible health benefits of consuming particular cultivars—*Hylocereus costaricensis*, in particular, has better nutritional qualities. In general, the study emphasizes how crucial it is to choose the right dragon fruit kinds in order to optimize dietary intake and advance general health and wellbeing.

## CONCLUSIONS

This research conducted a comprehensive analysis of nutritional components and phytochemical properties across dragon fruit cultivars, revealing significant differences in carbohydrate, protein, fat, moisture, fibre, ash, mineral content (including potassium, calcium, iron, zinc, magnesium, and phosphorus), anthocyanin, carotenoid, flavonoid, vitamin C, soluble sugar, and antioxidant activity. *Hylocereus costaricensis* is notable for its superior nutritional properties, including its mineral content; anthocyanin, carotenoid, flavonoid, and vitamin C contents; and antioxidant activity. These findings highlight the variety of cultivars, offering insights to consumers, nutritionists, and food scientists. Choosing the right cultivars is critical for meeting dietary needs and maximizing health benefits, and understanding the factors that influence nutritional profiles helps with product development and dietary planning for functional foods.

## ACKNOWLEDGEMENTS

The Head of the Discipline of Agrotechnology at Khulna University in Khulna, Bangladesh, is acknowledged by the authors for providing the required laboratory equipment.

### Funding

This research was funded by Khulna Agricultural University, Khulna 9100, Bangladesh. The study was also supported by Researchers Supporting Project number (RSP2024R283), King Saud University, Riyadh, Saudi Arabia. The funders had no role in study design, data collection and analysis, decision to publish, or preparation of the manuscript.

### Grant Disclosures

The following grant information was disclosed by the authors:
Khulna Agricultural University, Khulna 9100, Bangladesh.
Researchers Supporting Project number (RSP2024R283), King Saud University, Riyadh, Saudi Arabia.

### Competing Interests

The authors declare there are no competing interests.

### Author Contributions

- Afsana Yasmin conceived and designed the experiments, performed the experiments, analyzed the data, prepared figures and/or tables, authored or reviewed drafts of the article, and approved the final draft.
- Mousumi Jahan Sumi conceived and designed the experiments, performed the experiments, prepared figures and/or tables, authored or reviewed drafts of the article, and approved the final draft.
- Keya Akter conceived and designed the experiments, performed the experiments, prepared figures and/or tables, authored or reviewed drafts of the article, and approved the final draft.
- Rakibul Hasan Md. Rabbi conceived and designed the experiments, performed the experiments, prepared figures and/or tables, authored or reviewed drafts of the article, and approved the final draft.
- Hesham S. Almoallim analyzed the data, prepared figures and/or tables, authored or reviewed drafts of the article, and approved the final draft.
- Mohammad Javed Ansari analyzed the data, prepared figures and/or tables, authored or reviewed drafts of the article, and approved the final draft.
- Akbar Hossain analyzed the data, prepared figures and/or tables, authored or reviewed drafts of the article, and approved the final draft.
- Shahin Imran analyzed the data, prepared figures and/or tables, authored or reviewed drafts of the article, and approved the final draft.

### Data Availability

The raw data is available in the Supplemental File.

## Supplemental Information

Supplemental information for this article can be found online at http://dx.doi.org/10.7717/peerj.17719#supplemental-information.

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
