# Peer review of "Comparative analysis of nutrient composition and antioxidant activity in three dragon fruit cultivars"

_PeerJ, doi:10.7717/peerj.17719_

## Round 0.1 · original submission · Major Revisions

Please provide a comprehensively revised version addressing the editorial comments and a detailed rebuttal letter.

Reviewer 1 ·

Basic reporting

Figures are relevant, high quality, and well
labelled & described.

Experimental design

Methods described with sufficient detail &
information to replicate.

Validity of the findings

Impact and novelty not assessed.
Meaningful replication encouraged, where
rationale & benefit to literature are clearly
stated.

Additional comments

Overall, the authors have provided a regular, nutrient- and antioxidant-rich manuscript that, however, is in need of minor improvements before it could be considered for publication.

Annotated reviews are not available for download in order to protect the identity of reviewers who chose to remain anonymous.

Reviewer 2 ·

Basic reporting

Manuscript titled “Comparative analysis of antioxidant content and nutrient composition in three dragon fruit cultivars” reports a characterization of three different dragon fruit cultivars. There are some comments and suggestions for the authors:

1. In line 77, the phrase “Red-fleshed dragon fruit's high Fe concentration raises hemoglobin levels” is misleading. It appears that your intention is to correlate iron intake with proper hemoglobin levels, however, the sentence should be better expressed; please consider rephrasing.

2. In line 78, “…dietary fibre for diabetic individuals…”, it is unclear why would fiber be only for diabetic individuals; please consider rephrasing.

3. Line 262 mentions “white and yellow dragon fruit cultivars”. Fruits have been referred to by their scientific name until this point; please be consistent in how samples are named throughout the manuscript.

4. In line 407, the phrase “Conversely, the white and purple” is incomplete.

5. The manuscript cites various older papers. Please consider citing more recent ones whenever possible.

Experimental design

6. The method for flavonoid determination references the work of Close & McArthur, 2002, however, this paper is a review and does not contain any quantification method. Please confirm what technique was actually used and include the proper reference.

7. Line 303 mentions “SC50”, however, its definition or how it was calculated is not clearly stated in materials and methods, please add it. Same comment for correlation analysis in line 320.

Validity of the findings

8. Line 368 states that “The findings of this investigation align with those of Singh et al.”. Please elaborate on how your results are similar.

---

## Round 0.2 · accepted · Accept

Thanks for addressing all the reviewer comments.

Reviewer 2 ·

Basic reporting

See specific comments.

Experimental design

See specific comments.

Validity of the findings

See specific comments.

Additional comments

Manuscript titled “Comparative analysis of nutrient composition and antioxidant activity in three dragon fruit cultivars” was submitted after being initially revised. Comments and suggestions made by the present reviewer include:

1. Rephrasing an association made between iron present in dragon fruit and hemoglobin concentration. The change was made.

2. Similarly, a statement regarding diabetes and fiber was edited.

3. Completing an incomplete sentence. This was amended.

4. Using recent references where possible. Additional, more recent references have been cited.

5. Confirming the method used and correct citation for flavonoid determination. Another reference was reported, however, that reference describes methods based on chromatography (thin layer and liquid), but not gravimetric. The year and edition also appear to be incorrect, since the first edition is from 1984 (https://link.springer.com/book/10.1007/978-94-009-5570-7) unless I am mistaking this reference for a similar one, please confirm. Regardless of the reference cited, 80 % methanol will extract flavonoids, but it will also extract other alcohol-soluble substances, thus, a gravimetric method will most likely overestimate the samples’ flavonoid content.

6. Specifying the method used to calculate SC50 and correlation analyses. These have now been specified.

7. Providing additional information regarding the data reported by Singh and your own findings. Additional data is now mentioned.

According to the aforementioned changes made by the authors, it is apparent that most suggested changes were adequately addressed. Confirming the reference in comment 5 is the only issue that persists, however, this does not warrant another round of revision. There are no additional comments or suggestions for the present version of the document.